# Arbuscular Mycorrhizal Fungi Colonization of *Jatropha curcas* Roots and Its Impact on Growth and Survival under Greenhouse-Induced Hydric Stress

**Laura Yesenia Solís-Ramos** [1,*], **Antonio Andrade-Torres** [2,*], **Martin Hassan Polo-Marcial** [2], **Marysol Romero-Ceciliano** [1], **Cristofer Coto López** [1], **Carlos Ávila-Arias** [3] **and Keilor Rojas-Jiménez** [4]

1 Biotecnología de Plantas y Hongos Micorrícicos Arbusculares (Biotec-PYHMA), Escuela de Biología y Centro de Investigación en Biodiversidad y Ecología Tropical (CIBET), Universidad de Costa Rica, San Pedro de Montes de Oca, San José 11501-2060, Costa Rica; marysol.romero@ucr.ac.cr (M.R.-C.)

2 Biotecnología y Ecología de Organismos Simbióticos, CAUV-173 Ecología y Manejo de la Biodiversidad, INBIOTECA (Instituto de Biotecnología y Ecología Aplicada), Universidad Veracruzana, Av. de las Culturas Veracruzanas No. 101, Col. E. Zapata, Xalapa 91090, Veracruz, Mexico

3 Programa de Doctorado en Ciencias Naturales para el Desarrollo, Universidad Nacional, Tecnológico de Costa Rica y Universidad Estatal a Distancia, Heredia 86-3000, Costa Rica; cavila@utn.ac.cr

4 Laboratorio de Genética y Ecología de Microorganismos, Escuela de Biología, Universidad de Costa Rica, San Pedro de Montes de Oca, San José 11501-2060, Costa Rica; keilor.rojas@ucr.ac.cr

\* Correspondence: laura.solisramos@ucr.ac.cr (L.Y.S.-R.); aandrade@uv.mx (A.A.-T.)

**Abstract:** Arbuscular mycorrhizal fungi (AMF) provide benefits to host plants by enhancing nutrition and overall fitness. In this study, AMF species were isolated from the soil rhizosphere of *Jatropha curcas* and were identified and evaluated for their potential in fostering the development of *Jatropha* seedlings within a controlled greenhouse environment. The first experiment assessed the interplay between hydric stress and AMF inoculation on mycorrhizal colonization. The next experiment examined the impact of quercetin on mycorrhizal colonization. Out of 204 glomerospores corresponding to 28 species spanning 10 genera, *Acaulospora* (14) and *Scutellospora* (5) were the most abundant taxa. Six new records of AMF for Costa Rica are reported. Mycorrhizal colonization was observed in 43.2% of *Jatropha* plants (34.7% by AMF typical hyphae; arbuscules 8.9%; coils 5.6%; and vesicles 5.4%). Significant survival effects due to AMF inoculation under hydric stress were observed. On day 85, non-mycorrhizal plants subjected to hydric stress showed a mere 30% survival rate, whereas their mycorrhizal counterparts under hydric stress exhibited survival rates of 80% and 100% with and without irrigation, respectively. Furthermore, plants with irrigation and mycorrhizas showed greater hydric stress tolerance and superior growth. The inoculated plants, irrespective of irrigation, demonstrated mycorrhizal colonization rates of 63% and 72%, respectively. Quercetin did not affect *Jatropha*'s growth, but there were differences in AMF root colonization. In summary, these findings accentuate the viability of a native consortium in augmenting *Jatropha* survival, warranting consideration as a potent biofertilizer within greenhouse settings. The AMF described can be used for *Jatropha* propagation programs.

**Keywords:** bioinoculant; physic nut; flavonoids; *Jatropha*; symbiosis; water stress

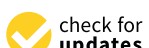



## 1. Introduction

Over the past decade, *Jatropha curcas* has emerged as a multipurpose plant, progressively harnessed for biofuel production and the extraction of compounds that hold significance across pharmacological, agricultural, and industrial domains. Its utility extends to revitalizing marginalized lands and disused mining sites [1–4]. With wide cultivation spanning countries such as Brazil, India, Mexico, Nicaragua, and Thailand [5–9], the species' multifaceted potential is acknowledged. Regrettably, a dearth of established technological protocols impedes the efficacious establishment of *Jatropha* crops.

Compelling evidence underscores the positive influence of arbuscular mycorrhizal fungi (AMF) on both plant growth and the microbial populations within the rhizosphere. This role is significant in enabling successful establishment under limited soil conditions and in abandoned mining sites [10]. Arbuscular mycorrhizal fungi (AMF) are cosmopolitan symbionts found in 80% of terrestrial plants [11,12], and assume a paramount role in sustainable agriculture by augmenting the uptake of water and vital nutrients, notably nitrogen (80%) and phosphorus (90%). The benefits of AMF encompass heightened drought resistance [13,14] and amelioration of soil-borne pathogens [15–17], along with substantive contributions to the remediation of contaminated soils [18]. Notably, consortia of native AMF exhibit superior effectiveness [19] compared to assemblages comprising exotic or solitary species [20].

The use of AMF presents a viable strategy to reduce losses incurred during multiplication, acclimatization, and adaptation of different plant species to other agroecological conditions [21]. In this context, the cultivation of forest plant species using AMF biotechnology could lead to higher profitability for producers, marked by substantial reductions in expenditures attributed to pest mitigation and chemical rectification of soil-based nutritional deficiencies [22].

The establishment of a symbiotic association between plants and AMF involves a complex signal exchange between the two entities [23]. For example, flavonoids stand as emblematic instances of such signals, comprising a class of secondary plant metabolites that exerts a pre-symbiotic influence on various aspects, including growth, spore germination, length, hyphal ramification, and formation of auxiliary cells and secondary spores [24].

The efficacy of AMF inoculation in plants depends on the capacity of mycorrhizal fungi to rapidly develop in soil with native species [25]. Successful AMF colonization strictly relies on the vitality of spores existing in the soil, often necessitating prolonged germination periods [25]. Notably, certain investigations reveal the role of flavonoids in modulating root colonization at an early stage of the symbiosis [24]. In addition, the introduction of quercetin into the soil has exhibited a propensity to increase AMF colonization, indicating that quercetin may be a key chemical signal that stimulates AMF associations [26]. Furthermore, higher concentrations of flavonoids in close proximity to roots have demonstrated the potential to foster spore germination and prompt the evolution of colonizing structures and runner hyphae while concurrently repressing the formation of spores and highly branched absorptive hyphal networks in the vicinity of roots [27].

The expanding global reach of drought-impacted regions harms crop production and yield, with the prevailing hydric deficit being the abiotic stress that most limits the growth and development of plants [28]. In *Jatropha curcas*, compelling indications underline the role of AMF in augmenting resilience amid both biotic and abiotic stressors, particularly in the context of metal-induced and saline stress [4,29,30].

Nevertheless, numerous of these studies have been conducted employing indigenous fungal species, often without explicit identification of the specific species contained within the inocula [31–33], or focusing on isolated AMF spores that colonize other species, such as *Casuarina equisetifolia* L. (*Glomus fasiculatum* and *Scutellospora calospora*) [34], or harnessing AMF species commonly encountered in stress-prone habitats such as saline soils [29]. However, introducing exotic AMF species into the diversity of native fungal species associated with local plant communities carries the potential for diverse impacts spanning positive, neutral, or negative outcomes on ecosystem functionality [35].

Our research group focused on identifying mycorrhizal associations in forest species, which is the case of *Jatropha curcas*, aiming to offer an alternative avenue to increase the success of reforestation and restoration programs for degraded lands. Thus, the objective of the present study was to isolate and characterize AMF richness in the rhizosphere of *Jatropha*, as well as to determine the effect of AMF (spores, hyphae, and inoculated root fragments) on germination and resistance to hydric stress with and without the external application of a flavonoid.

## 2. Materials and Methods

### 2.1. Plant Material and Arbuscular Mycorrhiza Inoculum

*Jatropha* seeds were obtained from the Fabio Baudrit Moreno Agricultural Experimental Station (EEAFBM, for its acronym in Spanish) at the University of Costa Rica. The samples were collected from the rhizosphere of *Jatropha* shrubs without the influence of surrounding vegetation in the northern part of Costa Rica in Balsa de Atenas, Alajuela (9°56'8.28" N, 84°22'37.51" W). The samples were collected at one meter from the base of the stem of the shrubs. Four subsamples were obtained using a 40 cm metallic cylinder at 40 cm of depth according to each cardinal point. The roots were separated from the extracted samples and placed in 70% alcohol, and the soil was dried in the shade at room temperature.

### 2.2. Glomerospore Isolation and Identification

Glomerospores were isolated using the method of wet sieving and decanting [36] with 60% sucrose solution (mass/v) [37]. One hundred grams of soil from each sample was weighed, deposited in one liter of running water, and stirred for one minute. The content was then passed through a series of sieves (37–1000 µm), and the residue of the last and second to last sieves (37–120 µm) was centrifuged with water at 1118 g (relative centrifugal force) (RCF = $(RPM)^2 \times 1.118 \times 10^{-5} \times r$) for 5 min. The supernatant was discarded and a 60% sucrose solution was subsequently added. The whole sample was then resuspended, mixed by inversion, and centrifuged at 280 g (RCF = $(RPM)^2 \times 1.118 \times 10^{-5} \times r$) for 1 min. The supernatant was passed through 120 and 230 µm sieves, and the residue from both sieves was transferred to a Petri dish using a wash bottle for immediate extraction and subsequent classification.

The samples were measured using a polarized light microscope (OLYMPUS CX43 binocular microscope with 48–75 mm interpupillary distance, 30° inclination angle, and 360-degree rotation) with an ocular ruler. The glomerospores were mounted on polyvinyl alcohol–lacto–glycerin (PVLG) [38], a mixture of PVLG and Melzer's reagent (1:1 *v/v*), for examination, as recommended by the International Culture Collection of (Vesicular) Arbuscular Mycorrhizal Fungi [39]. Based on the denomination of *Glomerospores* [40], morphological identification and classification were carried out according to established methods [37,41] and other relevant taxonomic criteria. The generic name *Rizhoglomus* [42] was used in the present study instead of *Rhizophagus*. The spores were maintained in suspension in distilled water in Eppendorf tubes at 10 °C until further analysis.

### 2.3. Clearing, Staining, and Colonization Estimation

The roots were mixed before processing to obtain a random sample. Twenty centimeters of secondary roots were washed to remove adhering organic matter. They were then processed with the method proposed by [43] with modifications. They were cleared with potassium hydroxide (KOH) at 10% in a water bath at 60 °C for 20 min. After this time, the KOH was removed and acidified with synthetic vinegar at room temperature for 24 h. The roots were stained with 0.05% trypan blue in a water bath at 60 °C for 30 min. The mycorrhizal and endophytic structures were observed using the method proposed by [44], where the roots were horizontally arranged on a microscope slide with two drops of water and observed under a compound microscope at 40× and 100× magnification to estimate the colonization percentage using the following formula:

$$\text{Colonization percentage} = \left[ \frac{Total\ number\ of\ colonized\ fields}{Total\ number\ of\ observed\ fields} \right] \times 100$$

The preparations of spores and stained roots were deposited as reference material in the Laboratory of Biotechnology of plants and arbuscular mycorrhizal fungi of the School of Biology, University of Costa Rica.

Indexes for Measuring the Community Structure

Simpson's dominance index: Predicts the probability that two consecutive samples taken at random will result in two individuals of the same species. It is expressed by the following formula:

$$\lambda = \Sigma p_i^2$$

where $p_i$ = abundance of species i, that is, the number of individuals of species *i* divided by the total number of individuals in the sample.

Shannon–Wiener index: based on the equitability of a particular ecosystem, it predicts to what species an individual randomly selected from a community would correspond [45]; it is commonly used to measure diversity using values between 0 (a single species) and S (total number of species with the same number of individuals) [46]. It is summarized in the following formula:

$$H' = -\Sigma p_i \ln p_i$$

Rarefaction index [47,48]: It uses non-parametric estimators that only require presence or absence data.

Chao 2. estimator with a lower degree of error in small samples.

$$\text{Chao}_2 = S + \frac{L^2}{2M}$$

where:

S = number of species;
L = number of species occurring in a single sample;
M = number of species occurring in two samples.

Jackknife 1: focuses on species that only occur in one sample.

Species accumulation curves: A graphical representation was constructed in Estimate S 9 for Windows using 100 randomized runs [49].

### 2.4. Molecular and Phylogenetic Analysis of Roots with Arbuscular Mycorrhizas

DNA was extracted from colonized roots to verify the presence of fungi and determine their taxonomy. For this purpose, approximately 200 mg of infected roots were cut and superficially disinfected with 95% ethanol for 2 min and 1% sodium hypochlorite for 5 min and washed five times with sterile water. The tissues were macerated and the DNA was subsequently extracted with the extraction kit protocol NucleoSpin Plant II, MACHERY-NAGEL (Duren, Germany) following the manufacturer's instructions. The 18S rRNA gene was amplified with primers NS5-ITS4 [50]. For the PCR reaction, Phusion High-Fidelity DNA Polymerase (New England BioLabs, Ipswich, MA, USA) was used with the following conditions: 94 °C for 3 min, 32 cycles at 94 °C for 30 s, 50 °C for 30 s, 72 °C for 1 min, and a final extension at 72 °C for 10 min. PCR products were checked in 1% agarose gels and then sequenced with an Applied Biosystems 3130 genetic analyzer (San Pedro, Costa Rica). Sequences were assembled using BioEdit v7.7.1 [51].

For the phylogenetic analysis, representative sequences of genera of the orders Glomerales and Diversisporales were retrieved from GenBank. The taxonomy of these sequences was carefully curated using Index Fungorum (http://www.indexfungorum.org/names/names.asp, accessed on 12 January 2023). Sequences were aligned in MAFFT v7 [52] and refined manually using BioEdit v7.7.1 [51]. The alignment used for subsequent phylogenetic analyses included 17 taxa and 480 positions of the 18S rRNA. A maximum-likelihood phylogenetic tree was constructed in FastTree 2.1.9 [53] using a GTR+G+I evolution model. The resulting tree was visualized and edited in MEGA7 [54].

### 2.5. Inoculum Production from Trap Crops

Trap crops were established as proposed by International Culture Collection of Vesicular–Arbuscular Mycorrhizal Fungi, United States (INVAM). Trap crops were es-

tablished using *Jatropha* soil (1.5 kg) mixed with sterilized soil in a ratio of 1:1 (*w/w*). The mixture (3 kg) was then transferred to plastic containers (20 cm diameter × 17 cm height) with *Zea mays* seeds and kept in a greenhouse for 4 months. The glomerospores were isolated according to the method specified in the section "Glomerospore extraction and identification". The supernatant was passed through a 37 μm sieve and the residue was transferred to a Petri dish using a wash bottle in order to extract the spores that were used as inoculum in Experiment 1.

### 2.5.1. Experiment 1: Effect of AMF on Germinated Seedlings under Hydric Stress

The experiment consisted of the following treatments: T1: NI − AMF, T2: WW − AMF, T3: NI + AMF, and T4: WW + AMF. Where NI: without irrigation throughout the experiment, WW: with irrigation at field capacity once a week, +AMF: with the application of arbuscular mycorrhizal fungi, and −AMF: without arbuscular mycorrhizal fungi. Two seeds were placed per *Jatropha* bag and planted at a depth of 3 cm with ten replicates per treatment. The bags were 25 cm in diameter and 30 cm in height and contained 4 kg of soil. After ten days of germination, the seeds were inoculated with AMF at a rate of 1 mL of aqueous suspension with 40 spores per replicate for treatments 3 and 4, which consisted of a consortium of 3 morphotypes (36, 17, and 747 spores of morphotypes 2, 3, and 1, respectively) isolated from the *Jatropha* trap crops. Sterilized soil (120 °C for 60 min) was used for this experiment. Data were recorded every 28 days until day 140.

### 2.5.2. Experiment 2: Impact of a Flavonoid Solution on Growth, Survival, and Arbuscular Mycorrhizal Fungi (AMF) Development in Jatropha

This experiment evaluated the effect of flavonoids (quercetin) on spore germination, hyphae growth ability, and colonization percentage of AMF inoculated into *Jatropha* roots. The experiment consisted of the following treatments: T1: 2 μM Q + AMF, T2: 5 μM Q + AMF, T3: 10 μM Q + AMF, T4: 10 μM Q + AMF in native soil (only treatment without sterile soil), T5: −Q + AMF, and T6: −Q − AMF. Where: Q: quercetin, −Q: without quercetin, +AMF: with arbuscular mycorrhizal fungi, and −AMF: without arbuscular mycorrhizal fungi. One hundred milliliters of distilled water was used for T5. Germinated shoots were inoculated with 3 g of fine roots from the *Jatropha* rhizosphere from Balsa de Atenas. One shoot was placed per bag with approximately 1300 g of a mixture of soil, vermicompost, and perlite at 2:1:1, respectively. For the flavonoid solution, 60.8 mg of quercetin (equivalent to 4.02 mmol/L) was weighed and dissolved in 50 mL of 95% ethanol, resulting in a stock solution of 100 μM of quercetin (SIGMA-ALDRICH, MW 302.24 anhydrous basis). The necessary amount of stock solution was dissolved in distilled water according to the desired concentration. A 100 mL part of quercetin was applied directly to the roots and the rest of the soil on day zero. On day 21, after inoculation, 100 mL of quercetin suspension or distilled water was applied again as appropriate. The quercetin was dissolved in 95% ethanol to obtain the stock and intermediate dilutions, and distilled water was subsequently used to obtain the dilutions of the respective concentrations. Five replicates per treatment were performed.

All experiments were carried out in the greenhouse of the School of Biology of the University of Costa Rica, San Pedro, Costa Rica, at a temperature of 23 to 31 °C.

### 2.6. Plant Growth Measurements

Plant survival was recorded four months after the seeds germinated in all treatment levels (experiments 1 and 2). The roots were washed to remove soil particles and subsamples were kept to evaluate fungal colonization. The following physical parameters were observed: germination (%), survival (%), height in cm (H), diameter at the base of the shoot in mm (DB), total fresh weight in g (FWS), total dry weight in g (DWS), root length in cm (RL), root width in mm (RW), fresh root weight in g (FRW), dry root weight in g (DRW), fresh leaf weight in g (FLW), dry leaf weight in g (DLW), leaf area, and mycorrhizal colonization percentage [43].

Mycorrhizal colonization was estimated using the method by Phillips and Hayman [43]. Root samples were randomized, clarified, and stained with trypan blue. Thirty 0.5 cm long root segments from each treatment were mounted on 50% glycerol and examined under a compound light microscope. The root pieces that contained one hypha or one or more vesicles or arbuscules were considered colonized. Colonization percentage was calculated as the proportion (%) of infected roots from the total of evaluated roots.

Mycorrhizal dependency (MD) or response to mycorrhizal colonization per plant in each treatment was calculated according to the following formula [55]:

$$\text{MD (\%) = dry weight of mycorrhized plant} - \frac{\text{mean dry weight of non-inoculated plant}}{\text{weight of mycorrhized plant}} \times 100$$

### 2.7. Experimental Design

Total glomerospore number, colonization percentage, and equitability and dominance indexes were analyzed with a Mann–Whitney U test [56].

A randomized complete block experimental design with 10 replicates per treatment was used for the first experiment and with 5 replicates for the second experiment. Analyses of variance (ANOVA) and Tukey tests were performed, both with a significance level of $p \leq 0.05$. The data were processed in SPSS for Windows (version 10.0). For the flavonoid experiment, the variables related to colonization were evaluated with a Kruskal–Wallis test. The variables that showed significant differences were analyzed with a Dunn's test in R Studio [57,58]. Assumptions of the ANOVA and Kruskal–Wallis test were verified with the 'stats' package version 3.5.1. The Dunn's test was performed with the 'dunn.test' package version 1.3.5 [57].

### 2.8. Soil Analysis

Soil analyses were carried out in the Soil Laboratory, CIA, UCR; the results are presented as Supplementary data.

## 3. Results

### 3.1. Mycorrhizal Colonization in the Soil Rhizosphere of Jatropha

Total mycorrhizal colonization of *Jatropha curcas* was quantified at 43.2%. The predominant fraction, 34.7%, was attributed to colonization by characteristic AMF hyphae, closely followed by arbuscules and coils at 8.9% and 5.6% colonization, respectively. There was 0.4% colonization by dark septate fungi expressed by septate hyphae (Figure 1).

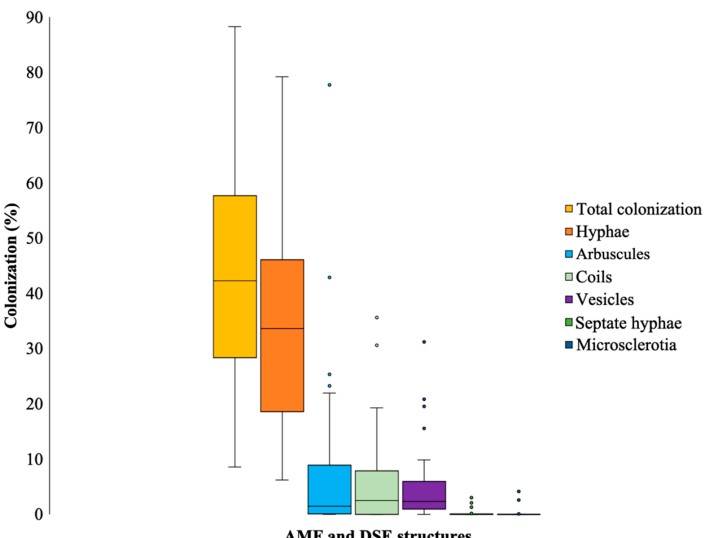

**Figure 1.** Total colonization and density of hyphae, arbuscules, coils, vesicles, and septate hyphae of AMF in the soil rhizosphere of *Jatropha curcas* collected in Costa Rica.

### 3.2. Richness and Abundance

A total of 204 glomerospores were isolated and characterized, which corresponded to a diverse range of 28 species spanning 10 genera (see Table 1 and Figure 2). Notably, the genus *Acaulospora* was the most abundant, with 12 distinct taxa, while *Scutellospora* presented 5 species. In terms of alpha diversity values, as evidenced by Simpson's reciprocal index (0.87) and Shannon's index (2.55), a low dominance was observed, since prevalence was attributed to only two species, where the species that showed the highest relative abundance within the AMF community structure were *Acaulospora rehmii* (0.209) and *A. scrobiculata* (0.209). Conversely, a subset of 11 taxa demonstrated limited presence within the community, each represented by a single individual. In light of the rarefaction index and considering the observed number of species (28), the non-parametric estimator Chao 1 forecasted a potential discovery of up to 55 species. Therefore, the species isolated directly from the field in this study constitute nearly 50% of the mycorrhizal community, according to the prediction.

**Table 1.** Richness and relative abundance of AMF associated with the rhizosphere of *Jatropha curcas* in Costa Rica. * New record of arbuscular mycorrhizal fungi for Costa Rica.

| Species | Relative Abundance |
| --- | --- |
| *Acaulospora rehmii* Sieverd. & S. Toro | 0.209 |
| *Acaulospora scrobiculata* Trappe | 0.209 |
| *Sclerocystis* sp. 1 | 0.124 |
| *Scutellospora* sp. 4 | 0.080 |
| *Acaulospora spinosa* C. Walker & Trappe | 0.070 |
| *Acaulospora* sp. 4 | 0.035 |
| *Acaulospora* sp. 2 | 0.030 |
| *Acaulospora* afin. *Splendida* | 0.025 |
| *Acaulospora tuberculata* Janos & Trappe | 0.025 |
| *Glomus* sp. 2 | 0.025 |
| *Rhizoglomus clarum* (T.H. Nicolson & N.C. Schenck) Sieverd., G.A. Silva & Oehl | 0.025 |
| *Scutellospora* sp. 1 | 0.020 |
| *Ambispora* sp. 1 * | 0.015 |
| *Acaulospora* sp. 3 | 0.015 |
| *Glomus* sp. 1 | 0.015 |
| *Sclerocystis coremioides* Berk. & Broome | 0.015 |
| *Acaulospora foveata* Trappe & Janos | 0.010 |
| *Acaulospora mellea* Spain & N.C. Schenck | 0.005 |
| *Acaulospora rugosa* J.B. Morton * | 0.005 |
| *Acaulospora* sp. 1 | 0.005 |
| *Funneliformis* sp. 1 | 0.005 |
| *Glomus taiwanense* (C.G. Wu & Z.C. Chen) R.T. Almeida & N.C. Schenck ex Y.J. Yao * | 0.005 |
| *Septoglomus* sp. 1 * | 0.005 |
| *Gigaspora margarita* W.N. Becker & I.R. Hall | 0.005 |
| *Scutellospora* sp. 2 | 0.005 |
| *Scutellospora* sp. 3 | 0.005 |
| *Scutellospora* afin. *fulgida* * | 0.005 |
| *Orbispora* sp. 1 * | 0.005 |

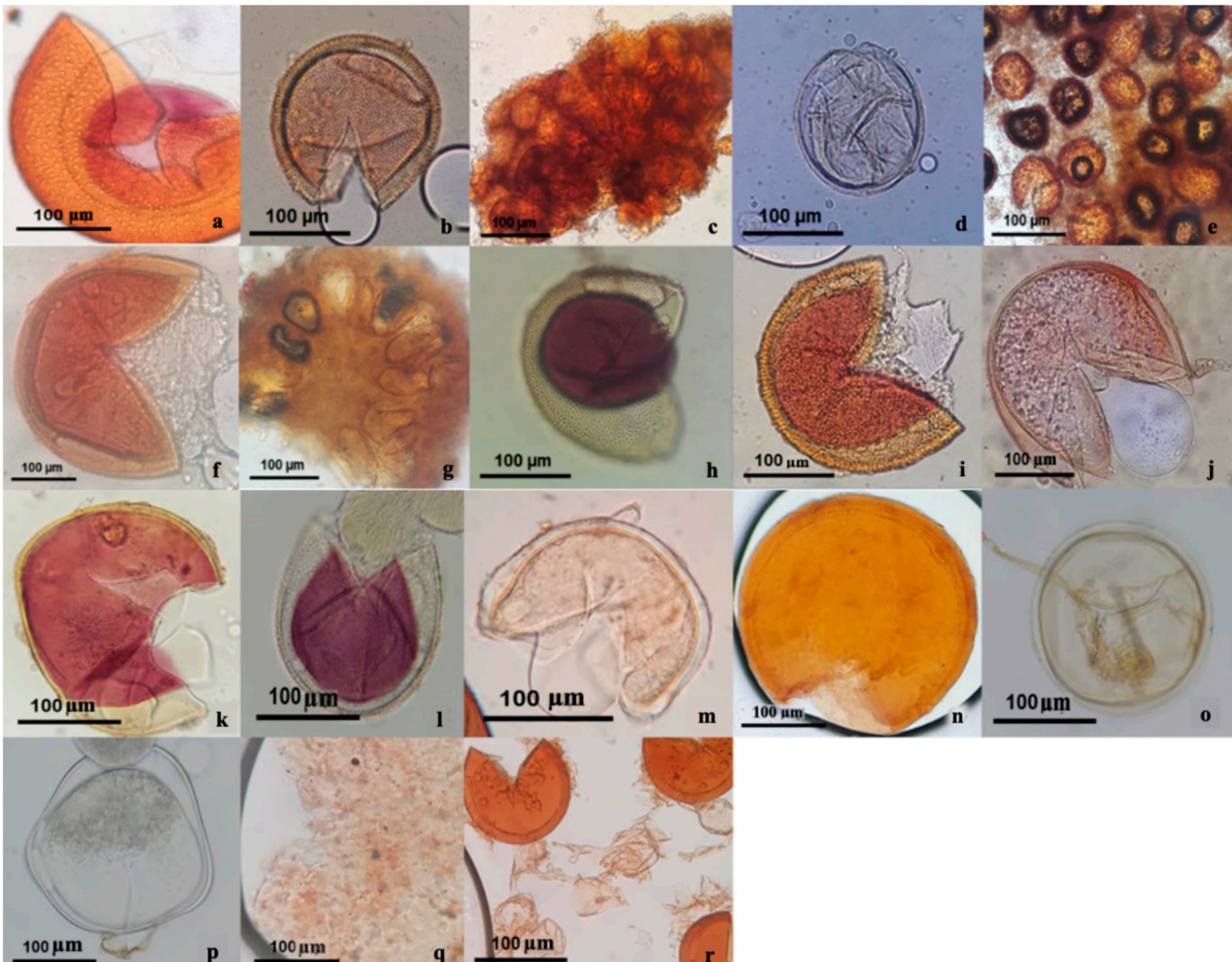

**Figure 2.** AMF community isolated from the soil rhizosphere of *Jatropha curcas*. (**a**) *Acaulospora foveata*. (**b**) *Acaulospora remhii*. (**c**) *Sclerocystis* sp. 1. (**d**) *Acaulospora* afin. *Splendida*. (**e**) *Glomus* sp. 2. (**f**) *Acaulospora spinosa*. (**g**) *Sclerocystis coremioides*. (**h**) *Acaulospora scrobiculata*. (**i**) *Acaulospora* sp. 4. (**j**) *Scutellospora* sp. 1. (**k**) *Scutellospora* afin. *Fulgida*. (**l**) *Acaulospora* sp. 1. (**m**) *Rhizoglomus clarum*. (**n**) *Acaulospora* sp. 2. (**o**) *Scutellospora* sp. 3. (**p**) *Scutellospora* sp. 4. (**q**) *Ambispora* sp. 1. (**r**) *Glomus* sp. 1.

### 3.3. Molecular and Phylogenetic Analysis of Infected Roots

The phylogenetic tree generated from the 18S rRNA sequences is consistent with the taxonomic classification of the taxa. The phylogenetic relationships between the genera *Glomus*, *Rhizoglomus*, and *Sclerocystis* were determined within the family Glomeraceae. With this marker, *Funneliformis* and *Septoglomus* were grouped together. It is important to highlight that the sequence obtained from the infected roots in the present study was classified within the genus *Rhizoglomus* (Figure 3).

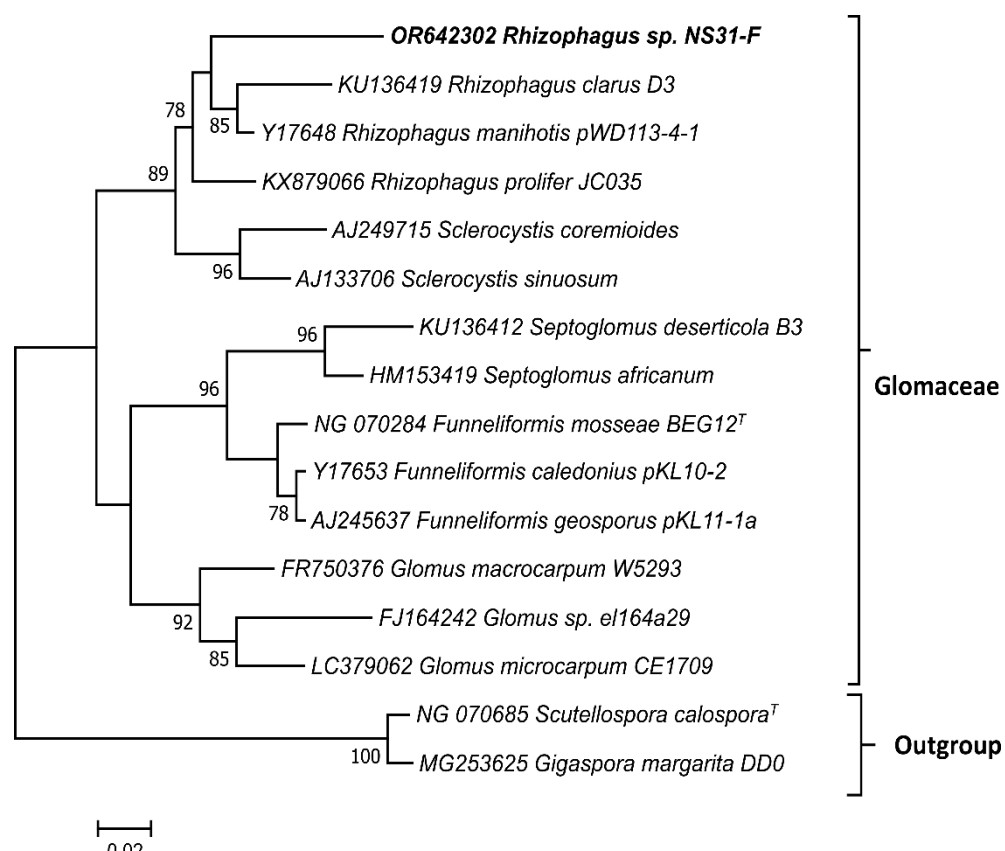

**Figure 3.** Maximum likelihood tree of selected genera of Glomaceae. The 18S rRNA sequences were aligned in MAFFT, the phylogenetic tree was generated in FastTree using a GTR+G+I model and visualized in MEGA 7. Bootstrap values are indicated in the nodes. The accession numbers, scientific names, and isolate names are also indicated. The sequence in bold, identified as *Rhizophagus* sp. NS31-F, was obtained in this study (accession number OR642302).

### 3.4. Effect of AMF on Germinated Seedlings under Hydric Stress

Significant differences were found in all the variables analyzed in the treatments with irrigation, both with and without mycorrhizas (T2 and T4), compared to those without irrigation, also with and without mycorrhizas (T1 and T3), during the evaluations conducted at 31 and 85 days (Table 2). This observation demonstrates that *Jatropha* shows higher growth with irrigation regardless of whether mycorrhizas are present. Furthermore, *J. curcas* exhibited resistance to water stress, as evidenced by plant survival in the greenhouse stage even under conditions where irrigation was absent, regardless of the concurrent presence of mycorrhizal symbiosis.

The plants subjected to hydric stress treatments exhibited heightened mortality rates, with the mycorrhizal-treated group displaying enhanced survival (50%) compared to the non-mycorrhizal counterparts (30%). Among treatments involving irrigation, a notable 100% survival was observed in mycorrhizal-treated plants, while their non-mycorrhizal counterparts exhibited an 80% survival rate. Thus, the presence of AMF increased the survival of *Jatropha* plants by a remarkable 20%, irrespective of irrigation status. It is noteworthy that despite mycorrhizal presence, 50% of total plants failed to survive in the absence of irrigation. Conversely, well-irrigated plants, regardless of mycorrhizal presence, displayed high survival rates (80%). *Jatropha* plants exhibited varying degrees of survival with or without irrigation during the greenhouse phase; however, their mortality was consistently decreased by 20% through mycorrhizal association.

Among the studied treatments, those without irrigation exhibited restrained growth across all variables, whereas the most pronounced growth was evident in treatments

benefiting from irrigation and mycorrhizal presence. A relationship between diameter at the base and plant height was identified, which indicates that the greater the height, the greater the diameter, which is explained by 93%. Notably, enlarged root length and width corresponded to heightened plant growth by 95% and 88%, respectively. An increase in root biomass (dry weight) correlated with increased plant growth by 63% and aerial biomass by 0.73%. In the case of fresh root weight (the whole root), there was higher aerial biomass (78%) and larger leaf area with higher weight in all treatments, with the highest values observed in the treatments with irrigation and mycorrhizas. This phenomenon underscores how mycorrhizal presence substantiates root development, enabling heightened nutrient and water absorption and higher plant growth. The increase in leaf area in plants with higher root biomass could explain a higher photosynthetic efficiency by increasing leaf nitrogen content, thereby further increasing dry weight, number of leaves, and leaf area. It is important to clarify, however, that root length was measured using the taproot, whereas mycorrhizas predominantly occupy secondary roots within the hyphal network.

Mycorrhizal inoculation yielded 63% and 72% colonization rates in non-irrigated and irrigated plants, respectively, whereas non-inoculated treatments exhibited no colonization. Among treatments supplemented with mycorrhizas, both non-irrigated and irrigated, there was a high percentage of hyphae (68.6% and 70.6%, respectively), arbuscules (T3 6.5% and T4 9%), and vesicles (T3 6.5% and T4 10.8%). These identified morphotypes were named 1, 2, and 3. From the NI + AMF treatment, 741 glomerospores were harvested, while 571 were obtained from the WW + AMF treatment. Within the mycorrhizal treatment deprived of irrigation, morphotype 1 (611 glomerospores) and morphotype 2 (130 glomerospores) were predominant. In contrast, the treatment involving AMF and irrigation yielded 463 glomerospores from morphotype 1 and 108 from morphotype 2.

**Table 2.** Experiment 1: Effect of AMF on *Jatropha curcas* germinated seedlings under hydric stress with or without mycorrhizas at 85 days. Mean values $\pm$ standard deviation.

| Treatment | NI − AMF (T1) | WW − AMF (T2) | NI + AMF (T3) | WW + AMF (T4) |
|---|---|---|---|---|
| **Variables** | | | | |
| H0 (cm) | 14.13 $\pm$ 1.45 [a] | 14.61 $\pm$ 1.98 [a] | 14.50 $\pm$ 1.99 [a] | 14.26 $\pm$ 0.96 [a] |
| H1 (cm) | 15.04 $\pm$ 1.48 [a] | 20.64 $\pm$ 3.99 [b] | 15.30 $\pm$ 1.94 [a] | 20.62 $\pm$ 1.38 [b] |
| H2 (cm) | 15.27 $\pm$ 1.69 [a] | 21.45 $\pm$ 3.11 [b] | 15.54 $\pm$ 1.90 [a] | 21.44 $\pm$ 1.98 [b] |
| DB0 (mm) | 2.29 $\pm$ 0.27 [a] | 2.31 $\pm$ 0.26 [a] | 2.25 $\pm$ 0.27 [a] | 2.15 $\pm$ 0.24 [a] |
| DB1 (mm) | 4.29 $\pm$ 1.47 [a] | 7.38 $\pm$ 1.69 [b] | 4.83 $\pm$ 1.06 [a] | 8.37 $\pm$ 0.46 [b] |
| DB2 (mm) | 4.36 $\pm$ 1.35 [a] | 9.88 $\pm$ 0.79 [b] | 5.88 $\pm$ 1.96 [a] | 11.10 $\pm$ 0.66 [b] |
| FWS (g) | 2.00 $\pm$ 0.83 [a] | 20.27 $\pm$ 4.30 [b] | 2.04 $\pm$ 0.65 [a] | 21.46 $\pm$ 2.96 [b] |
| DWS (g) | 0.46 $\pm$ 0.13 [a] | 4.34 $\pm$ 1.06 [b] | 0.51 $\pm$ 0.12 [a] | 5.48 $\pm$ 1.24 [b] |
| RL (cm) | 14.29 $\pm$ 2.42 [a] | 16.70 $\pm$ 2.38 [a] | 14.67 $\pm$ 2.04 [a] | 17.2 $\pm$ 2.03 [a] |
| RW (cm) | 6.54 $\pm$ 0.64 [a] | 7.96 $\pm$ 1.21 [b] | 3.84 $\pm$ 0.65 [bc] | 7.14 $\pm$ 0.93 [c] |
| FRW (g) | 0.47 $\pm$ 0.14 [a] | 5.70 $\pm$ 1.17 [b] | 1.24 $\pm$ 1.58 [a] | 7.15 $\pm$ 0.83 [b] |
| DRW (g) | 0.15 $\pm$ 0.04 [a] | 1.05 $\pm$ 0.27 [b] | 0.17 $\pm$ 0.05 [a] | 1.42 $\pm$ 0.17 [b] |
| FLW (g) | 0.79 $\pm$ 0.36 [a] | 7.50 $\pm$ 2.25 [b] | 0.39 $\pm$ 0.32 [a] | 8.25 $\pm$ 2.55 [b] |
| DLW (g) | 0.16 $\pm$ 0.07 [a] | 1.67 $\pm$ 0.56 [b] | 0.11 $\pm$ 0.07 [a] | 1.98 $\pm$ 0.55 [b] |

H0: height at the start of the experiment, H1: height after one month, H2: height after 85 days, DB0: diameter at the start of the experiment, DB1: diameter after one month, DB2: diameter after 85 days, FSW: total fresh stem weight, DSW: total dry stem weight, RL: root length, RW: root width, FRW: fresh root weight, DRW: dry root weight, FLW: fresh leaf weight, DLW: dry leaf weight. AMF: with arbuscular mycorrhizal fungi, −AMF: with arbuscular mycorrhizal fungi, NI no irrigation, WW irrigation. Numbers followed by different letters are statistically different ($p \leq 0.05$). ANOVA and Tukey tests with a significance level of $p \leq 0.05$ of the means of 10 replicates.

### 3.5. Impact of A Flavonoid Solution on Growth, Survival, and Arbuscular Mycorrhizal Fungi (AMF) Development in Jatropha

No significant differences were observed in any of the evaluated growth variables in the treatments involving mycorrhizal presence and in tandem with or without flavonoid application (Table 3). However, there was a marked variation in mean colonization rates between these treatments. The treatment with 10 μM quercetin and mycorrhizas in native soil exhibited the highest number of spores (69). Conversely, the treatment without quercetin and with mycorrhizas showed the highest percentage of hyphae (62%). In contrast, the non-inoculated roots showed a complete absence of any fungal structure (see Figure 4), confirming the efficacy of the sterilization method.

**Table 3.** Effect on growth and survival of *Jatropha* plants under different treatments at the root level, with or without arbuscular mycorrhiza and with or without flavonoids. Mean values ± standard error.

| Treatment | 2 μM Q +AMF | 5 μM Q +AMF | 10 μM Q +AMF | 10 μM Q +AMF (Native Soil) | −Q + AMF | −Q − AMF |
|---|---|---|---|---|---|---|
| **Variables** | | | | | | |
| TH (cm) | 14.8 ± 1.6 | 11.9 ± 0.7 | 11.1 ± 0.2 | 10.2 ± 0.2 | 14.2 ± 1.4 | 13.1 ± 1.2 |
| DB1 (mm) | 1.7 ± 0.3 | 1.0 ± 0.2 | 2.7 ± 1.8 | 1.0 ± 0.1 | 1.9 ± 0.3 | 1.6 ± 0.3 |
| FSW (g) | 30.5 ± 7.9 | 11.2 ± 5.0 | 4 ± 2.0 | 9.5 ± 1.0 | 38.5 ± 9.7 | 28.1 ± 7.4 |
| DSW (g) | 6.9 ± 2.4 | 1.7 ± 0.8 | 0.5 ± 0.3 | 2.0 ± 0.4 | 7.3 ± 2.1 | 6.0 ± 1.6 |
| LR (cm) | 13.3 ± 2.4 | 8.2 ± 4.0 | 6.6 ± 4.6 | 12.0 ± 2.6 | 15.4 ± 1.0 | 14.8 ± 3.1 |
| FRW (g) | 8.5 ± 2.4 | 2.8 ± 1.5 | 1.0 ± 0.9 | 1.9 ± 0.3 | 12.0 ± 3.3 | 9.1 ± 2.5 |
| DRW (g) | 1.8 ± 0.4 | 0.3 ± 0.3 | 0.2 ± 0.0 | 0.4 ± 0.1 | 1.3 ± 0.9 | 1.5 ± 0.5 |
| FLW (g) | 8.6 ± 3.0 | 3.1 ± 1.9 | 2.2 ± 2.2 | 0.8 ± 0.1 | 8.2 ± 1.6 | 6.4 ± 1.7 |
| DLW (g) | 1.3 ± 0.45 | 0.5 ± 0.29 | 0.3 ± 0.34 | 0.1 ± 0.03 | 1.3 ± 0.25 | 0.9 ± 0.26 |

Q: with quercetin, −Q: without quercetin, +AMF: with arbuscular mycorrhizal fungi, −AMF: without arbuscular mycorrhizal fungi, TH: total height, DB1: diameter at the base of the stem, FSW: fresh stem weight, DSW: dry shoot weight (g/plant), RL: root length, FRW: fresh root weight, DRW: dry root weight (g/plant), FLW: fresh leaf weight, DLW: dry leaf weight. No significant differences were observed.

Plants hosting arbuscular mycorrhizas exhibited an increased number of spores, hyphae, and vesicles across all treatments when compared with plants without mycorrhizal associations (as depicted in Figure 4). Nevertheless, the introduction of exogenous flavonoids in AMF-treated conditions did not consistently increase spore counts compared to AMF treatment without quercetin. Remarkably, the treatment involving 5 μM quercetin and AMF showed increased vesicle and arbuscule counts. The treatment of quercetin combined with re-inoculated AMF in native soil yielded a higher number of spores. Among the treatments with quercetin, the treatment with the highest quercetin concentration and mycorrhizas but without native soil showed the lowest number of spores, hyphae, and vesicles. Despite the differences, only number of spores ($\chi^2$ = 11,087, df = 5, $p$ = 0.049) and percentage of hyphae were statistically significant ($\chi^2$ = 11,849, df = 5, $p$ = 0.036).

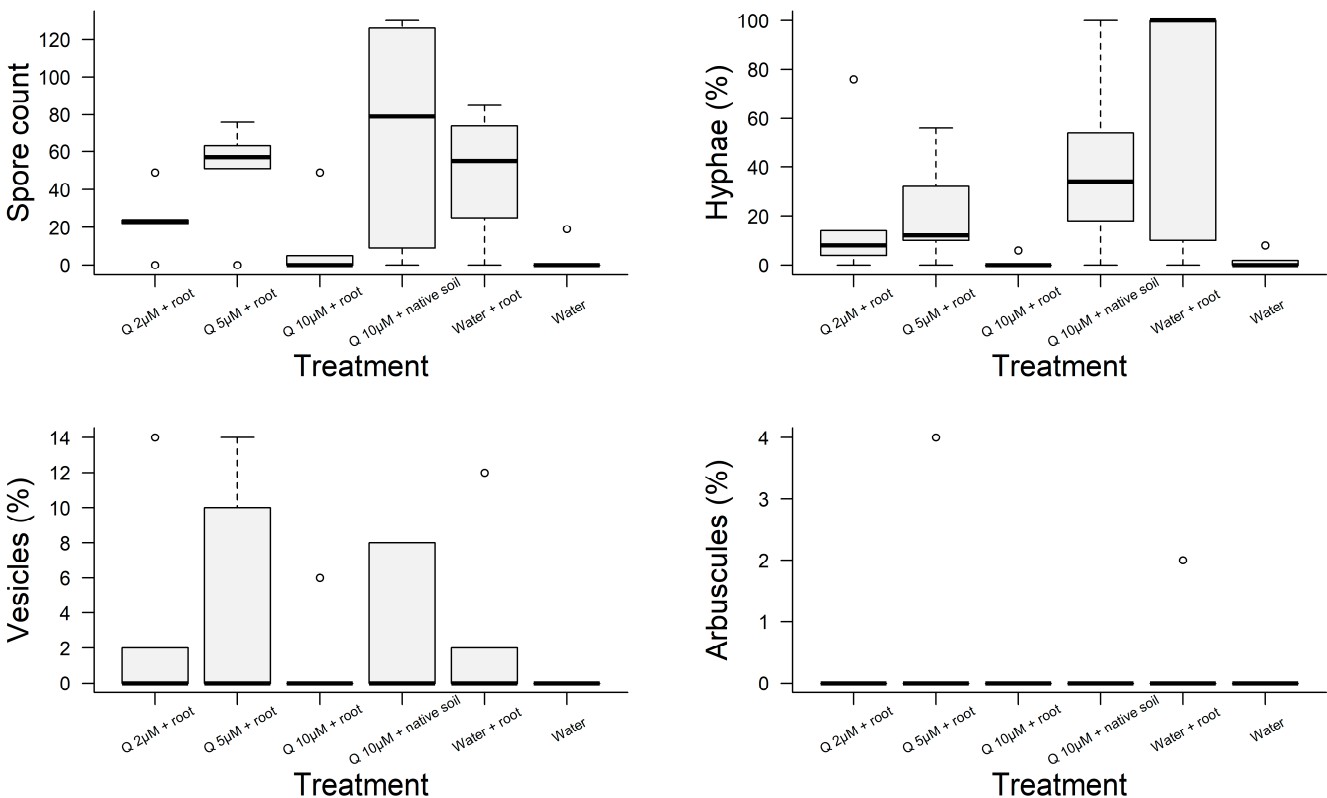

**Figure 4.** Box-Plots of AMF colonization of roots of *Jatropha curcas* plants under greenhouse conditions in treatments with or without mycorrhizas and with quercetin at different concentrations. No significant differences were observed for vesicles nor arbuscules. Only the number of spores ($\chi^2 = 11,087$, df = 5, $p = 0.049$) and percentage of hyphae were statistically significant ($\chi^2 = 11,849$, df = 5, $p = 0.036$).

## 4. Discussion

Members of the Euphorbiaceae family are recognized for their propensity to form mycorrhizal symbioses [59–63]. In the present study, a diverse assemblage of 28 AMF species spanning 10 distinct genera were successfully identified, with the dominance of *Acaulospora* (12 species), *Scutellospora* (5 species), and *Glomus* (3 species). *Acaulospora rehmii*, *A. scrobiculata*, and *Sclerocystis sinuosa* spp. were identified as the most abundant species. These results are similar to those reported by Charoenpakdee et al. [59,60], who identified 34 morphospecies of AMF, and the predominant genera were *Acaulospora* (16 spp.), *Glomus* (10 spp.), and *Scutellospora* (5 spp.), with *Acaulospora scrobiculata* being the most widely distributed species. A different study identified 10 species corresponding to five genera, with dominance of *Glomus* (3 spp.), *Rhizophagus* (=*Rhizoglomus*) (3 spp.), *Acaulospora* (2 spp.), and *Claroideoglomus* (1 spp.) [64]. Kamalvanshi et al. [62] reported the highest occurrence frequency for *G. intraradices* (100%), followed by *A. scrobiculata* (83%), *G. etunicatum* (50%), and *Glomus* 1 (50%). Similarly, Moreira et al. [63] recognized prevalent species such as *A. mellea* (74.4%), *A. morrowiae* (93%), and *Glomus* sp. 2 (72%), with nine species of the genus *Acaulospora* and nine species of *Glomus*. Together, these findings underscore *Jatropha curcas* as a mycorrhizal species, where the mycorrhizosphere predominantly features species from the Acaulosporaceae and Glomeraceae families [63]. Additionally, based on the present study, the family Gigasporaceae also contributes significantly to the composition of the mycorrhizosphere.

This study expands the number of AMF species known in Costa Rica, which highlights the importance of carrying out taxonomic and ecological studies in different hosts

and ecosystems, considering the possibility that the richness of AMF is greater than that obtained in this study [65].

The percentage of AMF colonization identified in the rhizosphere of *Jatropha curcas* was considered intermediate (43.2%). This categorization was substantiated by a higher presence of hyphae and a lower number of vesicles and arbuscules, which is consistent with other investigations involving the same species [63,66]. However, Jhan et al. [64] reported a high number of vesicles and arbuscules in *Jatropha*. The findings of the present study corroborate the notion that *Jatropha* is indeed reliant on mycorrhizal associations, displaying a moderate to high level of colonization, as previously established [59,60,62–64].

This study underscores the pivotal importance of taxonomically identifying AMF species associated with *Jatropha* and their contributory role in its growth, thereby paving the way for their utilization as viable bioproducts within both greenhouse and field contexts. Notably, certain commercially accessible AMF-based bioproducts have demonstrated limited efficacy, primarily due to the isolation of active organisms from environments that significantly differ from the conditions present in the target soils [67].

Drought stress is considered one of the most important abiotic factors limiting plant growth and yield [68] by affecting many metabolic activities due to reduced hydration under osmotic stress conditions [69]. Some plants have evolved mechanisms to mitigate abiotic stress through an increased root system or a symbiotic association with AMF. This symbiotic association leads to heightened water and nutrient absorption by the host, particularly under stress conditions [67,70,71]. In the case of hydric stress, the colonization of plant roots by AMF proves instrumental in enhancing water availability and bolstering tolerance to drought [72,73].

At the beginning of the present study, there were no reports regarding the role of AMF in *Jatropha curcas* within water-limited environments. This information is important because crops are established in regions characterized by prolonged dry seasons with a risk of droughts. Furthermore, its significance is accentuated in light of anticipated climatic variations. To address this knowledge gap, the effectiveness of native AMF isolated from the rhizosphere of *Jatropha* was evaluated in plants under hydric stress in the greenhouse stage in Costa Rica. This approach involved a consortium of indigenous species harvested from sites where the host species thrives. This methodology ensures that the AMF species exhibit heightened adaptability to the prevailing environmental conditions of the host sites, thereby enhancing their efficacy and infectivity in the target plant species [74]. The present study demonstrates that inoculating *Jatropha* plants with AMF, regardless of the presence of hydric stress, yields a 20% increase in plant survival compared to non-inoculated counterparts. This report agrees with previous studies showing increased survival in mycorrhized plants under hydric stress or saline stress [4,6].

However, the results reveal a scenario of colonization without discernible disparities in plant growth between specimens with and without AMF inoculation. Several factors, including soil sterilization, may contribute to this outcome by potentially eradicating the native microflora that facilitates nutrient uptake and fosters plant growth [32]. These results agree with another study that suggests two hypotheses to explain this response: firstly, *Jatropha* may possess a robust root structure endowed with abundant root hairs, thereby enabling enhanced nutrient uptake; alternatively, its substantial seed size may furnish sufficient reserves to support early-stage development [64]. This observation diverges from other investigations that underscore the reliance of plant species on AMF for fostering growth and heightening productivity [59,75], and can be thus used as amendments in crops not only for acclimatization but also for survival and growth success in plants under some abiotic stress [33,34,75–80].

The results of the present study did not show significant differences in the biomass of mycorrhized and non-mycorrhized *Jatropha* plants. One plausible explanation could be that biomass accumulation is a long-term process in plants, where short drought periods may not exert a significant impact [73]. However, some studies show higher dry weight in inoculated plants compared to non-inoculated plants due to an improvement in the ability

to absorb mineral nutrients, such as phosphorus, with the help of extraradical hyphae, and in the production of growth-promoting hormones, such as auxins and cytokinins [81].

The results also show a marked colonization percentage in treatments with mycorrhizas, whether subject to irrigation (70%) or non-irrigation (68%), as observed at 120 days post-cultivation. This observation aligns with the results reported by Pedranzani et al. [79], who documented a similar colonization rate of 60%. The temporal dynamics of colonization can indeed be influenced by various stressors experienced by plants, such as salinity [6]. Emerging evidence suggests that critical stages in the developmental cycle of AMF, encompassing processes like spore germination, colonization, extraradical hyphal elongation, and sporulation, may face hindrances under hydric stress [82]. This stress could potentially account for the observed limited presence of arbuscules and vesicles in the roots of inoculated *Jatropha*.

Upon harvest, two prevalent morphotypes were observed in the soil. In addition, a greater abundance of glomerospores was recorded in the non-irrigated treatment than in the irrigated one, implying a potential predilection of these species under hydric stress conditions. Diverse stressors have been shown to reduce AMF diversity and reshape the composition of AMF communities, resulting in assemblies dominated by phenotypically similar species that exhibit heightened tolerance to specific abiotic stresses [70]. Accordingly, it is necessary to assess fungal diversity to identify strains that can better promote plant development in response to escalating drought scenarios [67].

The survival and efficacy of AMF within the soil environment depend on the rapid colonization of host roots. This colonization is predominantly influenced by the competitive advantage held by native strains [25]. Recent investigations have underscored the contribution of flavonoids to regulating AMF symbiosis. The results of direct flavonoid application to spores and the interaction between plants and AMF are clearly functional responses. These results notably entail the stimulation of rapid spore germination and facilitation of root colonization by arbuscular mycorrhizas. Such facilitation, in turn, confers an advantage in resource acquisition and directly impacts developmental processes within the plant system [21]. The initial signals perceived by Glomeromycetes are those emanating from root exudates, wherein flavonoids play a pivotal role [83]. Among these flavonoids, quercetin merits special mention due to its remarkable ability to stimulate hyphal growth across diverse AMF genera [84–86]. These findings emphasize the intricate interplay between flavonoids and AMF development, shedding light on their multifaceted role within this symbiotic context.

The variables examined in this study did not show substantial differences when assessing shoot growth. Comparable outcomes, characterized by non-significant differences in plant growth parameters, have also been documented [64]. Variations in quercetin concentrations, as applied in different treatments, did not assume a decisive role in *Jatropha* growth. The comprehension of flavonoids as determinants of growth remains somewhat limited, and, thus, it is possible that they may not affect growth [87]. However, Fries et al. [88] observed a positive effect of the exogenous application of flavonoids, demonstrating enhanced mycorrhization and plant growth in clover and sorghum. It is noteworthy that these researchers employed higher quercetin concentrations. Therefore, the disparity in outcomes between their study and the present investigation could be attributed to the different quercetin dosages. The solvent used in the stock solution could also be a factor that influenced the absence of differences, since ethanol, even at very low concentrations, could be toxic depending on the type of system and plant [89]. Thus, this factor may have inadvertently contributed to the uniformity of results.

The treatments applied to the control group without mycorrhizas failed to manifest statistically significant differences. This result was unexpected since differences have been observed in most reports evaluating the effects of mycorrhizal fungi on *Jatropha curcas* [55,72]. One plausible explanation for this unexpected outcome may be attributed to an insufficient inoculum quantity, which could have hindered discernible growth variations. Disparities between treatment groups have been previously identified in experiments

involving *Jatropha* shoots with mycorrhizas, where the employment of a more substantial inoculum, either in terms of spore count or fine roots with hyphal colonization, has been pivotal [4,32,33,79,90]. This body of evidence underscores the critical importance of optimal inoculum levels in elucidating meaningful differences between treatment conditions and emphasizes the relevance of appropriate inoculation strategies.

In terms of glomerospore count, discernible differences were evident in treatments employing 5 and 10 μM quercetin concentrations, both with and without native soil, when compared with the control group without mycorrhizas. The treatment with 10 μM, AMF, and unsterilized native soil exhibited a conspicuous rise in glomerospore number, arguably attributed to the robust mycorrhizal synergy established with *Jatropha* [59,63]. In the case of percentage of hyphal colonization, there were also differences between the control and most of the experimental treatments, possibly due to the absence of inoculum and native soil in the control treatment to allow colonization. The treatments with the highest colonization percentage were those with intermediate (5 μM) and higher concentrations (10 μM) in native soil. Despite the presence of another treatment with the same concentration (10 μM), quercetin solution in the root affected plant survival and may have stressed the spore germination process [91]. It is pertinent to highlight that treatment 4 (T4) exhibited substantially superior colonization compared to treatment 3 (T3), where the variation seems traceable to the utilization of native soil. This assumption is underscored by the constancy in quercetin concentration across both treatments. Soil sterilization affects the composition and eliminates the spores that do not tolerate high temperatures [92].

The microorganisms present in the native soil could have allowed the plant to obtain higher resistance to the stress caused by the solution. It is well-established that mycorrhizal associations, under different types of stress, provide resistance to *Jatropha* plants [4,32,79]. The results from the treatments with non-sterile soil suggest that native AMF, along with other rhizosphere organisms, produce significant increases in growth, dry shoot and root weight, and mycorrhizal infection percentage compared to the control [93]. The homogeneous spatial distribution of mycorrhizal inocula in the soil may represent an additional factor influencing infection rates in unsterilized soil compared to sterilized and re-inoculated treatments [93]. Native AMF inoculated into tropical species has been more effective for plant growth and survival [93].

Despite the proposal of flavonoids like quercetin as potential promoters of spore germination [85,94–96], it is essential to acknowledge that instances yielding non-significant differences in germination outcomes have also been documented [97]. However, many germination experiments have been conducted in vitro and confined to specific species. Based on the results, a definitive link between quercetin and its influence on plant growth has not been established.

For example, Bécard et al. [98] conducted in vitro experiments and showed the role of different flavonoids in the symbiotic association. Intriguingly, their findings indicated instances where associations transpired even without flavonoids, yielding positive outcomes. This contradiction prompts us to question whether root-exuded flavonoids truly assume a pivotal role in establishing symbiosis in in vivo systems and their significance in real-world contexts.

The results presented in this study do not offer conclusive evidence, thereby precluding us from confirming a discernible impact of varying quercetin concentrations on growth or mycorrhization. Despite detecting variations in specific parameters like spore count and hyphal percentage, these differences proved insufficient to substantiate a causal relationship with quercetin concentration. To ascertain whether quercetin indeed promotes mycorrhization in in vivo systems of this nature, a more comprehensive array of investigations is required [85,94–96].

## 5. Conclusions

Understanding the facilitative role of arbuscular mycorrhizal fungi (AMF) in nutrient and water uptake in plants and in providing resilience against biotic and abiotic stressors

is a pivotal area warranting more in-depth investigation. Within the scope of this study, the findings underscore the critical significance of mycorrhizal symbiosis in *Jatropha curcas*. Through the present assessment, 28 species categorized into 10 genera were identified, reinforcing the breadth and complexity of this symbiotic relationship. In the context of the greenhouse-stage inoculation of *Jatropha curcas* with AMF, a notable 20% increase in survival was observed. This heightened survival rate endured irrespective of the presence or absence of hydric stress, emphasizing the consistently beneficial influence of AMF. The examination of quercetin treatments revealed that intermediate (5 μM) and elevated (10 μM) concentrations, when deployed in native soil, yielded higher colonization percentages, providing insights into potential avenues for optimizing mycorrhizal associations in practical applications. The present investigation further underscores the favorable impact of native AMF on *Jatropha* survival, and positions them as promising candidates for the development of potential tailored bioinoculants.

**Supplementary Materials:** The following supporting information can be downloaded at: https://www.mdpi.com/article/10.3390/agriculture13122197/s1, Table S1. Chemical analysis of the study sites. Table S2. Texture Analysis of the soils.

**Author Contributions:** Conceptualization, L.Y.S.-R.; Methodology, L.Y.S.-R., A.A.-T., M.R.-C., C.C.L. and C.Á.-A.; Formal analysis, L.Y.S.-R., M.H.P.-M., M.R.-C. and K.R.-J.; Investigation, L.Y.S.-R. and A.A.-T.; Writing—review & editing, L.Y.S.-R. and A.A.-T.; Supervision, L.Y.S.-R. and A.A.-T.; Project administration, L.Y.S.-R.; Funding acquisition, L.Y.S.-R. All authors have read and agreed to the published version of the manuscript.

**Funding:** This project was funded by project B6194 of the Vice-rectory for Research of the University of Costa Rica and the agreement between INBIOTECA-UV and the School of Biology-UCR. And The APC was funded by Vice-rectory for Research of the University of Costa Rica (L.Y.S.-R.) and the support of the HAPI-UV-2023 grant awarded (A.A.-T.).

**Institutional Review Board Statement:** Not applicable.

**Informed Consent Statement:** Not applicable.

**Data Availability Statement:** Not applicable.

**Acknowledgments:** We thank Guillermo Vargas from the E.E.F.B.M, UCR for providing the *Jatropha* seeds. This project was funded by project B6194 of the Vice-rectory for Research of the University of Costa Rica and the agreement between INBIOTECA-UV and the School of Biology-UCR. M.H.P.M. thanks the Mexican National Council for Science and Technology (CONACyT) for a mobility scholarship for a research visit at the School of Biology-UCR. Thanks to Krystell B. Poot de la Cruz, Andrea Ramírez Télles, Jenny Muñoz Valverde (Biology School), and Rolando Procupez (School of Chemistry) for their assistance. The support of the HAPI-UV-2023 grant awarded to AA-T is gratefully acknowledged. Thanks to Nidia González Lara and Helena Ajuria Ibarra for proofreading the English language.

**Conflicts of Interest:** The authors declare no conflict of interest.

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
