# Peer review of "Arbuscular Mycorrhizal Fungi Colonization of Jatropha curcas Roots and Its Impact on Growth and Survival under Greenhouse-Induced Hydric Stress"

_agriculture, doi:10.3390/agriculture13122197_

Round 1
Reviewer 1 Report
I appreciate the well written manuscript entitled " Biotechnological potential of arbuscular mycorrhizas in Jatropha curcas”. The manuscript describes a very interesting study. The authors did a lot of work. The subject is both interesting and worth publishing in “Agriculture”. That being said, the manuscript has the potential to be accepted. However, there is still some minor issues need to be addressed before the paper could be accepted attached to my report. Kind Regards.

Minor editing of English language required
Author Response
Dr. María Daniela Artigas Ramírez
Guest Editor
Special Issue "Beneficial Microorganisms and Crop Production"
Agriculture
In this letter we are sending our answers to the comments.
Comments |
Status |
Author´s answer |
Reviewer 1 |
|
|
Comments and Suggestions for Authors I appreciate the well written manuscript entitled " Biotechnological potential of arbuscular mycorrhizas in Jatropha curcas”. The manuscript describes a very interesting study. The authors did a lot of work. The subject is both interesting and worth publishing in “Agriculture”. That being said, the manuscript has the potential to be accepted. However, there is still some minor issues need to be addressed before the paper could be accepted attached to my report. Kind Regards.
|
|
|
1. I highly recommended that the author rework on the title to be more attractive in terms of what the readers will see new in this manuscript.
|
Agreed, applied |
New Title: Arbuscular mycorrhizal fungi colonization of Jatropha curcas roots and its impact on growth and survival under greenhouse-induced hydric stress.
|
2. Rephrase: Abstract: Certain plants exhibit higher colonization rates, improved growth, and enhanced survival when inoculated with native arbuscular mycorrhizal fungi (AMF).
|
Agreed, applied |
New phrase: Arbuscular mycorrhizal fungi (AMF) provide benefits to host plants by enhancing nutrition and overall fitness.
|
3. Replace rpm with “xg”. RCF=(RPM)2x1.118 x10-5 x r.
|
accepted - attended |
1118 xg (RCF = (RPM)2 x 1.118 x 10-5 x r) and 280 xg (RCF = (RPM)2 x 1.118 x 10-5 x r).
|
4. Why was not the method of Vierheilig with ink instead of toxic trypanblue used?
|
|
Because it is the standard protocol that we have established in the laboratory.
|
5. Italicize P throughout the manuscript
|
Agreed, applied.
|
|
6. Where are the letters of statistics
|
|
Not significant differences were observed.
|
7. a) spore count, b) percentage of hyphae, c) percentages of vesicles, d) percentage of arbuscules |
Agreed, applied in figure 4.
|
|
8. Too much citations |
Agreed |
The cites were revised.
|
|
|
|

Reviewer 2 Report
Comments Agriculture (2608047)
The manuscript titled “Biotechnological potential of arbuscular mycorrhizas in Jatropha curcas” is representing the interesting information and the author (s) performed a nice work. Overall, the manuscript is well structured; presenting novelty and authenticity of work. The results are reliable but for the improvements, some short comings in text are needed to be fixed for further improvement. Some important and serious flaws within the manuscript are given as:
It is really embarrassing to review the manuscript without line numbers.
Title: The present title is not highlighting/ showing relevancy to the present study, Need appropriate changing.
Abstract
Definitely, the authors performed this work but in writing/explaining do not used the words I, We, these must be changed throughout the manuscript. The language of manuscript is not scientific need to revise to improve the scientific soundness of article.
In Abstract line 2: Change the word extracted to Isolated
We isolated 204 glomerospores corresponding to 28 species spanning 10 genera change as: Out of 204 glomerospores,………………
Add future prospects in abstract.
Introduction heading is missing
Material and methods
Is there any Variety name of seeds that were used in the present study? What is EEFBM?
Section “Molecular and phylogenetic analysis of infected roots” what kind of infection the roots had?
Source of quercetin? And the method of its application?
The last paragraph of the section “Inoculum production from trap crops” and the Experiment II, Why two different soil (s) were used for experimentation? If so, these both have significant different physio-chemical properties before and after experimentations. Did the author(s) studies these properties as this study mainly deals with the root colonization of Jatropha curcas by AMF which is indeed a significant and novel work.
Results
Figure 1and 4: The quality of figure is not satisfactory (fig 4). There is no error bar/SD as well as statistical alphabets to differentiate the treatments. More, there is no Horizontal and vertical demonstration of values. All figures must be self explanatory. While in figure 4, need to add the detailed information of treatments instead of 1,2,3,4…
In Table 1, No mean+SD
Table 2: Make the letters/alphabets superscript.
Table 3: No statistical differentiation/alphabets
Need to remove the headings from discussion section.
Format of references in not in accordance with the style of journal.
Highlighted/enclosed in comments for the improvement of the Manuscript.
Author Response
Dr. María Daniela Artigas Ramírez
Guest Editor
Special Issue "Beneficial Microorganisms and Crop Production"
Agriculture
In this letter we are sending our answers to the comments.
Comments |
Status |
Author´s answer |
Reviewer 2: Comments and Suggestions for Authors The manuscript titled “Biotechnological potential of arbuscular mycorrhizas in Jatropha curcas” is representing the interesting information and the author (s) performed a nice work. Overall, the manuscript is well structured; presenting novelty and authenticity of work. The results are reliable but for the improvements, some short comings in text are needed to be fixed for further improvement. Some important and serious flaws within the manuscript are given as: It is really embarrassing to review the manuscript without line numbers.
|
|
|
1. Title: The present title is not highlighting/ showing relevancy to the present study, Need appropriate changing.
|
Agreed, applied. |
New Title: Arbuscular mycorrhizal fungi colonization of Jatropha curcas roots and its impact on growth and survival under greenhouse-induced hydric stress.
|
2. Abstract: Definitely, the authors performed this work but in writing/explaining do not used the words I, We, these must be changed throughout the manuscript. The language of manuscript is not scientific need to revise to improve the scientific soundness of article.
|
Agreed, applied. |
Revised. |
3. In Abstract line 2: Change the word extracted to Isolated
|
Agreed, applied.
|
Revised. extracted to isolated. |
4. We isolated 204 glomerospores corresponding to 28 species spanning 10 genera.
|
Agreed, applied. |
New phrase: 204 glomerospores were isolated,……………… |
5. Add future prospects in abstract.
|
Agreed, applied.
|
|
6. Introduction heading is missing
|
Agreed, applied.
|
|
7. Material and methods Is there any variety name of seeds that were used in the present study?
|
|
No, the seeds are identified as Jatropha seeds, there are no varieties.
|
8. What is EEFBM?
|
Agreed, applied.
|
Estación Experimental Agrícola Fabio Baudrit Moreno (EEFBM). |
9. Section “Molecular and phylogenetic analysis of infected roots” what kind of infection the roots had?
|
Agreed, applied
|
Molecular and phylogenetic analysis of roots with arbuscular mycorrhiza. |
10. Source of quercetin? And the method of its application?
|
Agreed, applied.
|
60.8 mg (equivalent to 4.02 mmol/L) were weighed and dissolved in 50 mL of 95% ethanol giving a stock solution of 100 µmol/L. From the stock solution the necessary amount was taken to dissolve in distilled water according to the desired concentration. A 100 mL part of quercetin was applied directly to the roots and the rest of the soil on day zero.
|
11. The last paragraph of the section “Inoculum production from trap crops” and the Experiment II, Why two different soil (s) were used for experimentation? If so, these both have significant different physio-chemical properties before and after experimentations. Did the author(s) studies these properties as this study mainly deals with the root colonization of Jatropha curcas by AMF which is indeed a significant and novel work.
|
|
Trap cultures are a standard technique to obtain the greatest richness of spores from a soil sample, so plants with high affinity are used to reproduce the scarce spores and to be able to detect them, in this case, trap cultures were established with corn plants inoculated with mycorrhizal fungi isolated from the rhizosphere of tempate. The objective of these is merely to increase the inoculum to establish experiments and here the spores with which the experiments themselves were worked were isolated. The soil used both in the trap cultures and in the experiments was sterile.
|
12. Results Figure 1and 4: The quality of figure is not satisfactory (fig 4). There is no error bar/SD as well as statistical alphabets to differentiate the treatments. More, there is no Horizontal and vertical demonstration of values. All figures must be self explanatory. While in figure 4, need to add the detailed information of treatments instead of 1,2,3,4…
|
|
The figures are presented without error bars because the method. There is only one data of colonization per treatment. Root samples were randomized, clarified, and stained with trypan blue. We mounted 30 root segments 0.5 cm long from each treatment on 50% glycerol and examined them under a compound light microscope. The root pieces that contained one hypha or one or more vesicles or arbuscules were considered colonized. We calculated colonization percentage as the proportion (%) of infected roots from the total of evaluated roots. Then, there is only one data of colonization per treatment. It is not posible to obtain error bars.
|
13. In Table 1, No mean+SD.
|
|
This table shows relative frequencies and are a function of the abundance of each species with respect to the total abundance (204 spores). Then, there is not posible to obtain SD.
|
14. Table 2: Make the letters/alphabets superscript.
|
Agreed, applied.
|
|
15. Table 3: No statistical differentiation/alphabets |
|
Not significant differences were observed. It is indicated in the text of the table. |
16. Need to remove the headings from discussion section. |
Agreed, applied.
|
|
17. Format of references in not in accordance with the style of journal.
|
Agreed, applied.
|
|
Comments on the Quality of English Language Highlighted/enclosed in comments for the improvement of the Manuscript.
|
Agreed, applied.
|
|

Reviewer 3 Report
The manuscript describes the diversity of arbuscular mycorrhizas in Jatropha curcas, effect of AMF on germination under hydric stress, and effect of a flavonoid solution on AMF development. The 28 species spanning 10 genera were isolated and identified. There are some issues that need to be addressed in the manuscript.
1. Figure 1, Figure 4, and Table 1 lack standard deviation.
2. In materials and methods, please clearly define E.E.F.B.M. of the University of Costa Rica.
3. In materials and methods, please check the sieves (120 y 37μm).
4. In materials and methods, please use ×g for centrifugation or indicate the instrument name.
5. In materials and methods, please check “a water bath at 120°C”.
6. Figure 2. Please check the unit, 100 mμ?? The AMF names should be italicized in the figure legend.
7. The 28 species were identified by 18S rRNA in Table 1? How to identify these species should be described. Or the sections of “Richness and abundance” and “Molecular and phylogenetic analysis of infected roots” should be merged.
8. The significant difference should be marked in Table 3.
9. The treatment with 10 μM quercetin and mycorrhizas in native soil exhibited the highest number of spores (69). 69%?
10. In Figure 4, what is the treatment from 1~6?
11. Table 3 lacks description, and Figure 4 is unclear. The section “Effect of a flavonoid solution on the development of AMF” should be modified.
12. The Supplementary data should be cited in the manuscript.
Author Response
Dr. María Daniela Artigas Ramírez
Guest Editor
Special Issue "Beneficial Microorganisms and Crop Production"
Agriculture
In this letter we are sending our answers to the comments.
Comments |
Status |
Author´s answer |
Reviewer 3. Comments and Suggestions for Authors The manuscript describes the diversity of arbuscular mycorrhizas in Jatropha curcas, effect of AMF on germination under hydric stress, and effect of a flavonoid solution on AMF development. The 28 species spanning 10 genera were isolated and identified. There are some issues that need to be addressed in the manuscript.
|
|
|
1. Figure 1, Figure 4, and Table 1 lack standard deviation.
|
|
The figures are presented without error bars/SD because the method. There is only one data of colonization per treatment. Root samples were randomized, clarified, and stained with trypan blue. We mounted 30 root segments 0.5 cm long from each treatment on 50% glycerol and examined them under a compound light microscope. The root pieces that contained one hypha or one or more vesicles or arbuscules were considered colonized. We calculated colonization percentage as the proportion (%) of infected roots from the total of evaluated roots. Then, there is only one data of colonization per treatment. It is not posible to obtain error bars/SD.
|
2. In materials and methods, please clearly define E.E.F.B.M. of the University of Costa Rica. |
Agreed, applied. |
Estación Experimental Agrícola Fabio Baudrit Moreno (EEFBM).
|
3. In materials and methods, please check the sieves (120 y 37μm).
|
Agreed, applied
|
corrected to 120 and 230 μm. |
4. In materials and methods, please use ×g for centrifugation or indicate the instrument name.
|
Agreed, applied |
1118 xg (RCF = (RPM)2 x 1.118 x 10-5 x r) and 280 xg (RCF = (RPM)2 x 1.118 x 10-5 x r).
|
5. In materials and methods, please check “a water bath at 120°C”.
|
Agreed, applied
|
Corrected to 60°C. |
6. Figure 2. Please check the unit, 100 mμ?? The AMF names should be italicized in the figure legend.
|
Agreed, applied |
Corrected to μm. |
The 28 species were identified by 18S rRNA in Table 1? Answer NO. How to identify these species should be described. Or the sections of “Richness and abundance” and “Molecular and phylogenetic analysis of infected roots” should be merged.
|
|
It is different. First Glomerospores were isolated from soil using the method of wet sieving and decanting (Gerdemann and Nicolson 1963), then with the isolated spores we identify and classify morphologically according to Oehl et al. (2011), Błaszkowski (2012) and additional taxa. This is a taxonomic method based in morphology, we have published papers about spores in soils. Table 1 is based in the results. The taxonomy of Glomeromycota is based on morphology of the glomerospores. The figure 3 corresponds to the molecular methods. This method is used to study the AMF found inside the roots, it is not based on morphology because the diagnostic characters are found only in spores, not in hypha, inside the roots, however, there are molecular markers to identify some genera or species of glomeromycota, then we study root samples with AMF colonization and the figure 3 was obtained.
In conclusion: DNA was extracted from roots of the 30 Jatropha trees collected, and only those that amplified were sent for sequencing. Table 1 was based on morphological identification from spores isolated from the rhizosphere. Richness and abundance and Molecular and phylogenetic analysis sections cannot be merged.
|
8. The significant difference should be marked in Table 3.
|
|
Not significant differences were observed. It is indicated in the text of the table
|
1. The treatment with 10 μM quercetin and mycorrhizas in native soil exhibited the highest number of spores (69). 69%?
|
Agreed |
It is 69 spores, see a, figure 4.
|
2. In Figure 4, what is the treatment from 1~6?
|
Agreed
|
Corrected in the figure 4. |
3. Table 3 lacks description
|
Agreed, corrected. |
We included as description: Effect on growth and survival of Jatropha plants under different treatments at root level, with or without arbuscular mycorrhiza and with or without flavonoids.
|
4. Figure 4 is unclear.
|
Agreed |
Corrected Figure 4. a) spore count, b) percentage of hyphae, c) percentages of vesicles, d) percentage of arbuscules.
|
5. The section “Effect of a flavonoid solution on the development of AMF” should be modified.
|
Agreed |
Corrected: "The impact of a flavonoid solution on the growth, survival, and arbuscular mycorrhizal fungi (AMF) development in Jatropha."
|
6. The Supplementary data should be cited in the manuscript.
|
Agreed,
|
corrected. Soil analysis Soil analyses were carried out in the Soil Laboratory, CIA, UCR, and the results are presented in Supplementary data.
|
